

# Atropos: specific, sensitive, and speedy trimming of sequencing reads

John P. Didion[1], Marcel Martin[2] and Francis S. Collins[1]

[1] National Human Genome Research Institute, National Institutes of Health, Bethesda, MD, USA
[2] Science for Life Laboratory, Department of Biochemistry and Biophysics, Stockholm University, Stockholm, Sweden

## ABSTRACT

A key step in the transformation of raw sequencing reads into biological insights is the trimming of adapter sequences and low-quality bases. Read trimming has been shown to increase the quality and reliability while decreasing the computational requirements of downstream analyses. Many read trimming software tools are available; however, no tool simultaneously provides the accuracy, computational efficiency, and feature set required to handle the types and volumes of data generated in modern sequencing-based experiments. Here we introduce Atropos and show that it trims reads with high sensitivity and specificity while maintaining leading-edge speed. Compared to other state-of-the-art read trimming tools, Atropos achieves significant increases in trimming accuracy while remaining competitive in execution times. Furthermore, Atropos maintains high accuracy even when trimming data with elevated rates of sequencing errors. The accuracy, high performance, and broad feature set offered by Atropos makes it an appropriate choice for the pre-processing of Illumina, ABI SOLiD, and other current-generation short-read sequencing datasets. Atropos is open source and free software written in Python (3.3+) and available at https://github.com/jdidion/atropos.

# INTRODUCTION

All current-generation sequencing technologies, including Illumina, ABI SOLiD, and Ion Torrent, require a library construction step that involves the introduction of short adapter sequences at the ends of the template DNA fragments. Depending on the sequencing platform and the fragment size distribution of the sequencing library, an often substantial fraction of reads will consist of both template and adapter sequences (Fig. 1A). Additionally, the error rates of these sequencing technologies vary from 0.1% on Illumina to 5% or more on long-read sequencing platforms. Error rates tend to be enriched at the ends of reads (where adapters are located), thus exacerbating the effects of adapter contamination. Adapter contamination and sequencing errors can lead to increased rates of misaligned and unaligned reads, which results in errors in downstream analysis including spurious variant calls (*Del Fabbro et al., 2013*; *Sturm, Schroeder & Bauer, 2016*). Certain sequencing protocols may introduce other artifacts in sequencing reads. For example, some methylation sequencing (Methyl-Seq) protocols result in artificially

Corresponding author
John P. Didion,
john.didion@nih.gov

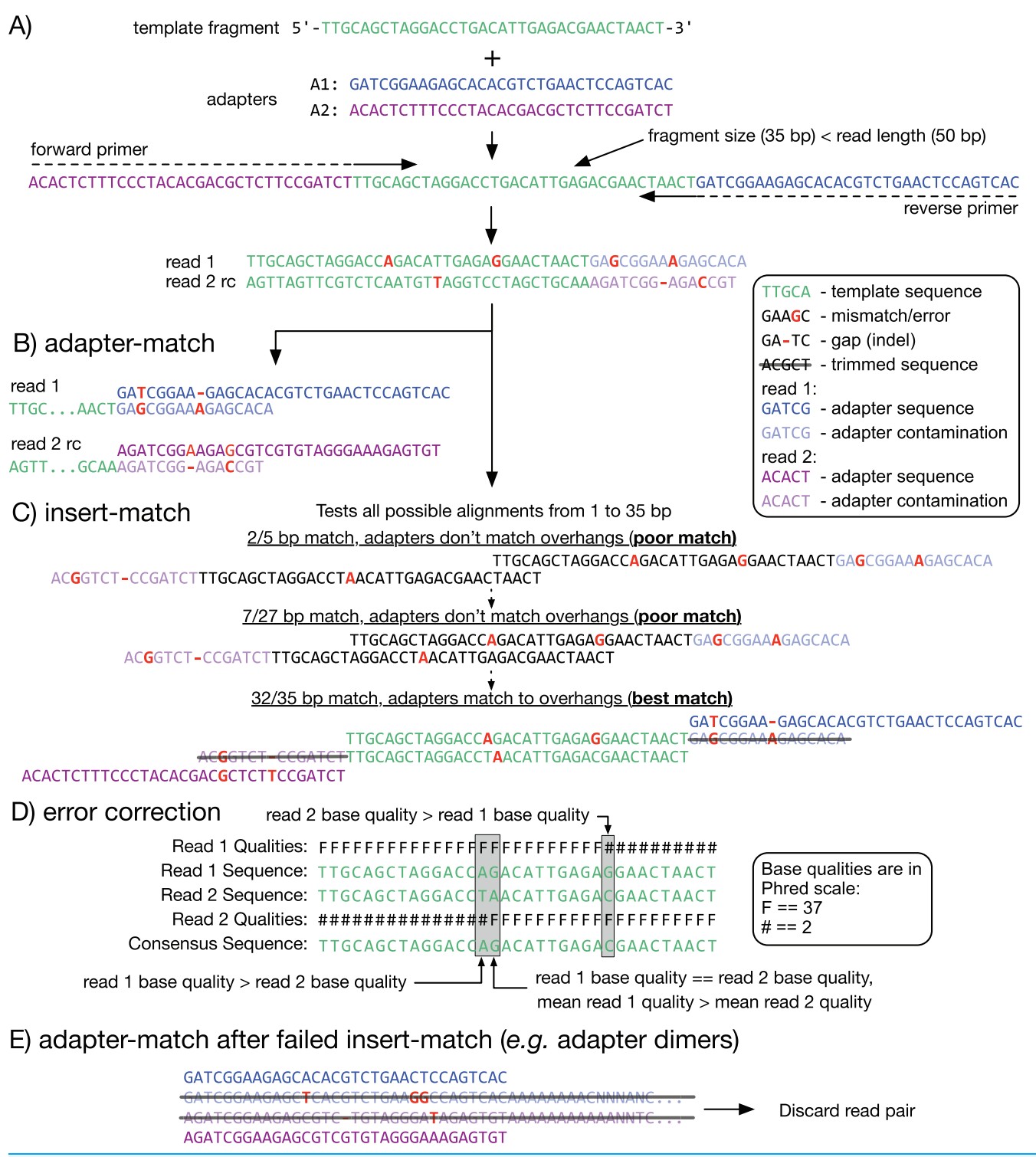

**Figure 1  Adapter detection and trimming.** (A) When a fragment (or insert; green) is shorter than the read length, the read sequence will contain partial to full-length adapter sequences (blue and purple). (B, C) Methods for detecting adapter contamination using semi-global alignment. Adapter-match (B) identifies the best alignment between each adapter and the end of its corresponding read. Insert-match (C) first identifies the best alignment between read 1 and the reverse-complement (rc) of read 2; if a valid alignment is found, then adapters are matched to the remaining overhangs. (D) If a match is found, the overlapping inserts can be used for mutual error correction. The consensus base is the one with the highest quality, or, if the bases have equal quality, the one from the read with highest mean quality. (E) If insert-match fails (for example, with an adapter dimer) adapter-match is performed. Reads that are too short after trimming are discarded.

methylated bases at the 3′ ends of reads that can lead to inflated estimates of methylation levels (*Bock, 2012*).

Read trimming is an important step in the analysis pipeline to mitigate the effects of adapter contamination, sequencing errors, and other artifacts. The development of tools for read trimming is an active area of bioinformatics research, thus there are currently many options. In terms of adapter trimming, these tools fall into two general categories: (1) those that rely solely on matching the adapter sequence (*adapter-match trimming*) using semi-global alignment (which is the only option available for single-end reads; Fig. 1B); and (2) those that leverage the overlap between paired-end reads to identify adapter starting positions (*insert-match trimming*; Fig. 1C) (*Sturm, Schroeder & Bauer, 2016*). Cutadapt (*Martin, 2011*) is a mature and feature-rich example of a tool that provides adapter-match trimming, while SeqPurge (*Sturm, Schroeder & Bauer, 2016*) is a recent example of a highly accurate insert-match trimmer designed specifically for paired-end data. Additionally, hybrid tools are available that optimize their choice of read trimming method based on the type of data. Skewer (*Jiang et al., 2014*) and AdapterRemoval (Version 2) (*Schubert, Lindgreen & Orlando, 2016*) are fast and accurate hybrid trimmers that work with both single-end and paired-end data. However, choosing a read-trimming tool currently requires a trade-off between feature set, efficiency, and accuracy. Furthermore, even state-of-the-art tools still have a relatively high rate of over-trimming (removing usable template bases from reads) and/or under-trimming (leaving low-quality and adapter-derived bases in the read sequence) (*Sturm, Schroeder & Bauer, 2016*).

We sought to develop a read-trimming tool that would combine the best aspects of currently available software to provide high speed and accuracy while also offering a rich feature set. To accomplish this aim, we used Cutadapt as a starting point, as it provides the broadest feature set of currently available tools and is published under the MIT license, which allows modification and improvement of the code. We focused on making three specific improvements to Cutadapt: (1) improve the accuracy of paired-end read trimming by implementing an insert-match algorithm; (2) improve the performance by adding multiprocessing support (as Cutadapt is currently only able to use a single processor); and (3) add important additional features such as automated trimming of Methyl-Seq reads, automated detection of adapter sequences in reads where the experimental protocols are not known to the analyst, estimation of sequencing error, and generation of quality control (QC) metrics. Because these modifications required substantial changes to the Cutadapt code base, and because there are software tools that depend on the current implementation of Cutadapt, we chose to "fork" the Cutadapt code base and develop our software, Atropos, as a separate tool. Here, we show that we have accomplished our three aims. In addition to extending the already rich set of features provided by the original Cutadapt tool, Atropos demonstrates paired-end read trimming accuracy that is superior to other state-of-the-art tools, and it is among the fastest read trimming tools when a moderate number of parallel execution threads are used (4). Furthermore, Atropos achieves a performance increase that is roughly linear with the number of threads used, making it the fastest tool when eight or more threads are available.

## MATERIALS AND METHODS

### Implementation

Atropos is developed in Python (3.3+) and is available to install from GitHub or via one of several package managers (see Data Availability).

#### *Semi-global alignment*

Traditionally, the overlap between two sequences is detected by computing an optimal semi-global alignment (*Gusfield, 1997*, Section 11.6.4), which is the same as global alignment except that neither initial nor trailing gaps are penalized. This allows the sequences to shift relative to each other. An optimal semi-global alignment maximizes the sum of alignment column scores, thus tending to favor longer over short overlaps. Since score-based optimization is often not intuitively understood, the adapter alignment algorithm uses edit operations instead, which has the advantage that it gives the user the ability to specify a "maximum error rate" as an intuitive parameter. For a given alignment between read and adapter, the error rate is computed as the number of edits (mismatches, insertions, deletions) divided by the length of the matching part of the adapter. Minimizing the edit distance while at the same time not penalizing end gaps would lead to optimal but meaningless zero-length overlaps; thus, a hybrid approach is chosen. The adapter alignment algorithm computes edit distances for all allowed shifts of the adapter relative to the read. Among those having an error rate not higher than the specified threshold, the shift (and therefore alignment) with the highest number of matches is chosen.

We summarize the algorithm here; see *Martin (2013)*, Section 2.2 for details. Let $a$ and $r$ be the nucleotide sequences of the adapter and sequencing read, respectively, and let $m = |a|$, $n = |r|$. Adapter alignment computes edit distances $D(i,j)$ between the $i$-length prefix of $a$ and the $j$-length prefix of $r$ for all $i = 0, \ldots, m$ and $j = 0, \ldots, n$ with the standard dynamic-programming (DP) recurrence

$$D(i, j) = \min\{D(i - 1, j - 1) + [a_i \neq r_j], D(i - 1, j), D(i, j - 1)\} \tag{1}$$

The base cases are $D(i,0) = 0$ or $D(i,0) = i$ and $D(0,j) = 0$ or $D(0,j) = j$, depending on the adapter type, allowing to skip a prefix of $a$ and/or $r$ at no cost. The algorithm additionally keeps track of $M(i, j)$, which is the number of matches between the prefixes of $a$ and $r$, and of the "origin" $O(i, j)$, which is the number of skipped characters in $r$ in the optimal alignment (if negative, characters in $a$ are skipped instead). All three DP matrices $D, M, O$ are filled in at the same time, after which the cells of the bottom row ($i = m$) are inspected. They represent possible end positions of the adapter sequence within the read. For each position $j$, the error rate is computed from $D(m, j)$ and $O(m, j)$, and positions with a too high error rate are discarded. If positions remain, the one with the highest number of matches $M(m, j)$ is returned as the position $J$ of the adapter sequence. Together with the start of the adapter sequence at $O(m, J)$, the adapter sequence can then be removed from the read.

Observing that no backtrace within the DP matrix is required, the actual implementation keeps only a single column of the matrices in memory for better cache locality. Significant runtime improvements are achieved by employing the optimization

described by *Ukkonen (1985)* of stopping the computation of a column as soon as the costs are too high and provably cannot decrease for the remainder of the column. When the user supplies an anchored adapter and disables insertions and deletions (indels) at the same time, the algorithm also switches to a much simpler variant that computes only the Hamming distance between the adapter and a prefix or suffix of the read.

### Insert match algorithm

For each read pair, the insert-match algorithm uses the same semi-global alignment algorithm described above (with indels disabled) to find all possible alignments between the first read and the reverse complement of the second read that satisfy specificity thresholds (Fig. 1C). Specificity is determined by the combination of up to three user-configurable thresholds: (1) minimum number of overlapping bases; (2) maximum number of mismatch bases; and (3) random mismatch probability (*Sturm, Schroeder & Bauer, 2016*). The probability of a random match at $k$ bases out of the $n$ bases being compared is computed using the binomial distribution:

$$P = \sum_{i=k}^{n} \frac{n!}{i!(n-i)!} p^i (1-p)^{n-i} \qquad (2)$$

The candidate alignments are tested in order of decreasing length until one is found in which the overhanging sequences on either end match the user-specified adapter sequences. Comparison between the adapter and overhang sequences is done using a constrained adapter-match approach. Briefly, starting at the end of the insert overlap, a pairwise comparison is made between the adapter and the read at each possible offset. The offset that best satisfies the user-configurable specificity thresholds (the same three described above) is taken to be the location of the adapter sequence, and all bases from that position to the 3′ end of the read are removed. If an adapter is only found in one of the two reads, then the same offset is used to trim both reads, under the assumption that the location of the adapter sequence must be symmetric across the read pair.

Optionally, the overlapping inserts can be used for mutual error correction (Fig. 1D). Where the aligned inserts have mismatches, the base with the highest quality score is chosen as the consensus. When the bases have equal quality, there is an option to leave the bases unchanged, convert them both to $N$, or to choose the base from the read with the highest mean quality as the consensus. There are additional options to (1) completely overwrite one read in the pair if its quality is very poor; and/or (2) merge the overlapping read pair into a single read, which avoids double-counting overlapping read pairs in read depth-based analyses.

If no insert match is found, or if an adapter is not found in an overhang, then an unconstrained adapter-match approach is attempted separately in each read (Fig. 1E).

### Parallel processing

The performance improvements in Atropos relative to Cutadapt and other read trimming tools are based in two observations: (1) each read (or read pair) is trimmed separately, and thus trimming can be parallelized across multiple processor cores; and (2) a significant

fraction of the execution time is spent decompressing input files and re-compressing results. Compression of sequencing data is increasingly becoming necessary due to the large volumes of data generated in sequencing experiments.

To address the first bottleneck, we implemented a parallel processing pipeline based on the Python multiprocessing module. Briefly, a single thread is dedicated to a "reader" process that loads reads (or read pairs) from input file(s), with support for a variety of data formats and automatic decompression of compressed data. Reads are loaded in batches, and each batch is added to an in-memory queue. A user-configurable number of "worker" threads (which is constrained by the number of processing cores available on the user's system) extract batches from the queue and perform trimming and filtering operations on the reads in the same manner as Cutadapt.

Atropos addresses the second bottleneck by offering a choice of three modes for writing the results to disk. The first two modes involve adding the results to a second in-memory queue, from which a dedicated "writer" process extracts batches and performs the serialized write operation. These two modes differ in how the trimmed reads are compressed—in worker-compression mode, each worker is responsible for compressing the results using the Python gzip module prior to placing the results on the queue, whereas in writer-compression mode, the writer process performs compression using the much faster system-level gzip program. The choice between these two modes is selected automatically based on the number of worker threads used, with worker-compression mode becoming faster than writer-compression mode work "typically" when at least eight threads are available. The third output mode, called "parallel writing," does not use a dedicated writer process (and thus an additional worker process can be used in its place). Instead, each worker process writes its results to a separate file. This can dramatically reduce the execution time of the program (50% reduction in our experiments; see Results) and is generally compatible with downstream analysis since many mapping and assembly tools accept multiple input files (and for those that don't, gzipped files can be safely concatenated without needing to be decompressed and recompressed). An additional speed-up is gained by recognizing that the reader process often finishes loading data well before the worker processes finish processing it; thus, an additional worker thread is started as soon as the reader process completes.

### Adapter detection

Often, details of sequencing library construction are not fully communicated from the individual or facility that generated the library to the individual(s) performing data analysis. For example, the majority of datasets in the NCBI Sequence Read Archive lack adapter sequence annotations. Manual determination of sequencing adapters and other potential library contaminants can be a tedious and error-prone task. Thus, we implemented in Atropos a command that automatically identifies adapters/contaminants from a sample of read sequences. First, a profile is built of $k$-mers (where $k$ is a fixed number of consecutive nucleotides, defaulting to $k = 12$) within $N$ read sequences (where $N$ defaults to 10,000). When at least eight consecutive A bases are detected, those bases along with all subsequent bases (in the 3′ direction) are first trimmed, as that

pattern is a strong indicator that the sequencer scanned past the end of the template (i.e., the length of the fragment + adapter is less than the read length; Fig. 1E). Additionally, low-complexity reads are excluded, where complexity $X(S)$ is defined as follows. Let $C(i,S)$ be the number of elements of a nucleotide sequence $S = s_1, \ldots, s_n$, that are nucleotide $i \in A,C,G,T.$

$$X(S) = - \sum \frac{C(i, S) \cdot \log(C(i, S))}{\log(2)} \tag{3}$$

Sequences with $X(S) < 1.0$ are defined as low-complexity. All remaining $k$-mers are counted, and each $k$-mer is linked to all of the sequences from which it originated. This process continues iteratively for increasing values of $k$, with only those read sequences linked to high-abundance $k$-mers in the previous iteration being used to build the $k$-mer profile in the next iteration. $k$-mer $K$ is considered high-abundance when:

$$|K| > \frac{N \cdot (l - k + 1) \cdot O}{4^k} \tag{4}$$

where $l$ is the read length and $O = 100$ by default. Finally, high-abundance $k$-mers of all lengths are merged to eliminate shorter sequences that are fully contained in longer sequences.

Atropos reports to the user an ordered list of up to 20 of the most likely contaminants. Because adapter sequences have been designed not to match any known sequence in nature, a sequence (or pair of sequences) that occurs at high frequency and matches a known adapter sequence is likely to be the true sequence(s) used as adapters in the dataset. Thus, our algorithm optionally matches the high-abundance $k$-mers to a list of known adapters/contaminants. We provide a list of commonly used adapter sequences, or the user can choose to supply their own. When a contaminant list is not provided, or when the adapter does not match a known sequence, we advise the user to take caution when using the results of this detection process, as a highly abundant sequence might simply be derived from a frequently repeated element in the genome.

### Error rate estimation

Quality and adapter trimming is sensitive to the choice of several parameters. For example, relative to datasets with typical rates of sequencing error, datasets with higher error-rates require higher thresholds for mismatches and/or random-match probability during insert- and adapter-matching to perform with the same level of sensitivity. Thus, we implemented in Atropos a command that provides an estimate of the error rate in each input file. The error command gives the choice between two algorithms: (1) averaging all base qualities across a sample of reads, which is fast but likely overestimates the true rate of sequencing error (*Dohm et al., 2008*; *DePristo et al., 2011*); and (2) the shadow regression method proposed by *Wang et al. (2012)*, which more accurately estimates error rates at the cost of reduced speed and greater memory usage.

### Quality control metrics

Examination of QC metrics is another important aspect of sequence analysis pipeline. For example, the widely used FastQC (*Andrews, 2010*) tool generates statistics such as per-sequence and per-base quality scores and GC content, sequence length distribution, sequence duplication levels, and frequency of potential contaminants. QC is commonly performed both before and after read trimming to identify any systematic data quality issues, to observe the improvements in data quality due to trimming, and to ensure that trimming does not introduce any unintended side-effects. Since both read trimming and QC involve iterating over all reads in the dataset, we reasoned that implementing both operations in the same tool would reduce the overall processing time, and also eliminate the need to install two separate tools. Thus, we implemented an option in Atropos to collect QC metrics before and/or after trimming.

Additionally, we implemented an Atropos module for MultiQC (*Ewels et al., 2016*), a program that generates nicely formatted reports from metrics output by a variety of bioinformatics tools for one to many samples. Given summary files generated by Atropos (one per sample, in JSON format), the MultiQC module will generate interactive versions of the same static plots offered by FastQC, as well as a summary table of the most important metrics.

### Shared Cutadapt and Atropos improvements

In addition to improvements in the semi-global alignment algorithm above, Atropos also benefits from the following improvements that were made to Cutadapt subsequent to the publication of *Martin (2011)*, but prior to the Atropos fork, and are therefore features in both programs.

- Adapters can now be *anchored*, which limits the read positions at which they will be matched. An anchored 5′ adapter thus matches only if it is a prefix of the read, and a 3′ adapter only if it is a suffix of the read. This is useful, for example, when one or both sequencing adapters are known to be ligated directly to a PCR primer.
- *Linked adapters* combine a 5′ with a 3′ adapter. Trimming multiple adapters from each read was also supported previously, but linked adapters make it possible to require that one of them is a 5′ adapter and one a 3′ one.
- *IUPAC ambiguity codes* are fully supported. Thus, adapter sequences containing characters such as N (matching any nucleotide), H (A, C, or T), Y (C or T) work as expected. They are useful when adapters contain barcodes or random nucleotides. The nucleotides and ambiguity codes are internally represented as patterns of four bits, in which each set bit corresponds to an allowed nucleotide. Comparisons are thus simple "binary and" operations, resulting in no runtime overhead.
- *Paired-end data* can be trimmed with sequences specified for the forward and reverse reads independently. Read pairs are guaranteed to remain in sync. Even *interleaved* data (paired-end reads in a single file) is accepted.
- Quality trimming can now work in a *NextSeq-specific* mode in which spurious runs of high-quality G nucleotides at the 3′ end of a read are correctly trimmed. NextSeq

instruments use "dark" or "black" cycles for G nucleotides, making them unable to distinguish between regular G and reaching the end of the fragment.

- Other additions include support for *trimming a fixed number of bases* from a read, support for files compressed using the bzip2 and lzma algorithms, and improved filtering options.

## Benchmarks

### Simulated data

We evaluated both the speed and the accuracy of Atropos relative to other state-of-the-art read trimming tools using both simulated and real-world data (Table 1). As trimming of single-end reads is unchanged from the original Cutadapt method and is also decreasing in relevance as most current experiments use paired-end data, we focused our benchmarking on trimming of paired-end reads. Sturm et al. demonstrate that Skewer (*Jiang et al., 2014*) and SeqPurge (*Sturm, Schroeder & Bauer, 2016*) stand out as having superior performance in paired-end read trimming, and Schubert et al. also demonstrate exceptional performance of AdapterRemoval (*Schubert, Lindgreen & Orlando, 2016*); thus, we chose to benchmark Atropos against these tools. We also compared the new insert-match algorithm against the adapter-match algorithm that is used by Cutadapt, and which can be enabled in Atropos using the "--aligner" command line option.

To simulate paired-end read data, we use the ART simulator (*Huang et al., 2012*) that was modified by Jiang et al. to add adapter sequences to the ends of simulated fragments. ART simulates reads based on empirically derived quality profiles specific to each sequencing platform. A quality profile consists of distributions of quality scores for each nucleotide at each read position, expressed as read counts aggregated from multiple sequencing experiments, where quality scores are in Phred scale ($-10\log_{10}(e)$, where $e$ is the probability that the base call is erroneous). We developed an $R$ script to artificially inflate the error rates in an ART profile to a user-defined level. For each row in the profile with quality score bins $e_1..e_n$ and corresponding read counts $r_1..r_n$, the overall error rate can be computed as:

$$E = \frac{\sum_{i=1}^{n} e_i r_i}{\sum_{i=1}^{n} r_i} \tag{5}$$

We use the $R$ function *optim* with the variable metric ("BFGS") algorithm to optimize a function that adds an equal number of counts $C$ to each Phred-score bin in the distribution and then compares the overall error rate to the user-desired error rate $E'$:

$$f(C, E') = \frac{\sum_{i=1}^{n} e_i (r_i + C)}{\sum_{i=1}^{n} (r_i + C)} - E' \tag{6}$$

We simulated ~800k 125 bp paired-end reads using the Illumina 2500 profile at error rates that were low/typical (~0.2%, the unmodified profile), intermediate (~0.6%), and high (~1.2%). We evaluated the accuracy of the tools on the simulated data by comparing each trimmed read pair to the known template sequence. We counted the frequency of following outcomes: the fragment does not contain adapters but is trimmed anyway ("wrongly trimmed"), or the fragment does contain adapters but either too few bases or

**Table 1 Description of data sets.**

| Dataset | Error rate (%) | Read length | Total read pairs | Reads w/Adapters | Adapter bases |
|---|---|---|---|---|---|
| Simulated 1 | 0.20 | 125 | 781,923 | 325,982 | 12,447,262 |
| Simulated 2 | 0.60 | 125 | 780,899 | 325,105 | 12,416,861 |
| Simulated 3 | 1.20 | 125 | 782,237 | 325,860 | 12,464,235 |
| GM12878 WGBS* | 2.79 | 125 | 1,000,000 | 57,130 | 3,082,003 |
| K562 mRNA-seq* | 4.31 | 75 | 6,100,265 | 14,384 | 749,451 |

Note:
For the real data sets, * actual error rates are unknown—we estimate error rates from base qualities over a sample of 10,000 read pairs; and total adapter content is unknown—we provide the number of reads containing exact matches for the first 35 adapter bases, and the number of adapter bases present.

too many bases were removed ("under-trimmed" or "over-trimmed"). We also counted the total number of under- and over-trimmed bases.

### Real data

We also benchmarked the tools on two real-world datasets. First, we sampled ~1 M read pairs from a whole-genome bisulfite sequencing (WGBS) library generated from the GM12878 cell line. Second, we used 6.1 M paired-end mRNA-seq reads generated from the K562 cell line. Both of these datasets were generated by the ENCODE project (*ENCODE Project Consortium, 2012*). Since the genomic origins of the templates are not known a priori, we instead compared the read trimming tools based on improvement in the results of mapping the trimmed versus untrimmed reads. We used STAR (*Dobin et al., 2013*) to map the mRNA-seq reads to GRCh38, and we used bwa-meth (*Pedersen et al., 2014*) to map the WGBS reads to the bisulfite-converted GRCh38. We also compared the results of only adapter trimming to the results of adapter trimming plus additional quality trimming using a minimum quality threshold of 20 (Phred-scale).

One characteristic of the mRNA-Seq dataset is that average read 2 quality is substantially lower than read 1 (estimation by the "atropos error" subcommand: 6.7% versus 2.0%). In practice, when encountering a read pair in which one end is of much lower quality than the other, the Skewer algorithm essentially overwrites the former with the later, leading to more precise alignment. Atropos provides a specific option for this case ("--w"), which we make use of in our benchmark in order to fairly compare Atropos with Skewer. However, this gives these tools a perhaps unfair advantage over AdapterRemoval and SeqPurge which do not have such an option.

### Computing environments

Although sequence analysis is sometimes performed using a desktop computer, analysis of the volumes of data currently being generated increasingly requires the use of high-performance computing facilities ("clusters"). The hardware architecture of a cluster is often different from that of a desktop computer. Most importantly, storage in a cluster is typically centralized and accessed by the compute nodes via high-speed networking. Such an architecture inevitably adds latency to file reading and writing operations ("I/O"). Cluster nodes also typically have more processing cores and memory available than

desktop computers. This means that the performance of software with intensive I/O usage (such as read trimming) is likely to be quite different on a desktop versus a cluster. To examine the impact of these architectural differences, we ran the benchmarks for simulated data on both a desktop computer (a Mac Pro) having a 3.7 GHz quad-core Intel Xeon E5 processor and 32 GB RAM, and on a cluster node having 64 2.4 GHz Intel Xeon E5 cores and 256 GB memory, and with all data being read from and written to network-accessible storage over a 1 Gbit ethernet connection.

### Reproducibility and reusability

With increasing importance being placed on both the reproducibility of results in scientific publications and the reusability of software and pipelines, we endeavored to provide a benchmark workflow that can be easily executed and extended by anyone with access to modern computing resources.

First, we "containerized" all of the software tools used in this paper—including trimming tools, read mapping tools, and supplementary tools used to evaluate results and generate tables and figures (Table S1). We also created minimal containers for all of the data used in this paper—including benchmark datasets, reference genomes, annotation databases, and indexes used by the mapping tools. Specifically, we created Docker (*Boettiger, 2015*) image specifications ("Dockerfiles"), generated the images, and uploaded them to a public repository on the Docker Hub (see Data Availability).

Second, we implemented our benchmark workflow using the Nextflow (*Di Tommaso et al., 2017*) framework. Importantly, Nextflow enables workflows to be run either locally or in most cluster environments, and supports running containerized software via either a Docker or Singularity (*Kurtzer, 2016*) client (depending on the operating system).

Instructions for running our workflow, along with all of the source scripts, are available in our GitHub repository (see Data Availability).

## RESULTS

### Simulated data

### Performance

On a desktop computer with four processing cores, we found that AdapterRemoval had the fastest overall execution time, followed closely by SeqPurge, Atropos (in parallel write mode), and Skewer (Fig. 2A; Table S2).

As expected, execution times on a cluster node using four threads were approximately 20% greater than those observed on a desktop computer (Fig. 2B; Table S3). We expect that much of this disparity is due to the increased latency involved in network-based I/O on the cluster, although some may also be explained by CPU differences (3.7 GHz Intel on the desktop versus 2.4 GHz on the cluster node).

When increasing the number of parallel execution threads from 4 to 8 and 16, Atropos achieves a roughly linear decrease in execution time. Interestingly, the execution times of AdapterRemoval, SeqPurge, and Skewer do not substantially decrease when increasing the number of the threads from 8 to 16. With 8 and 16 threads, Atropos using the

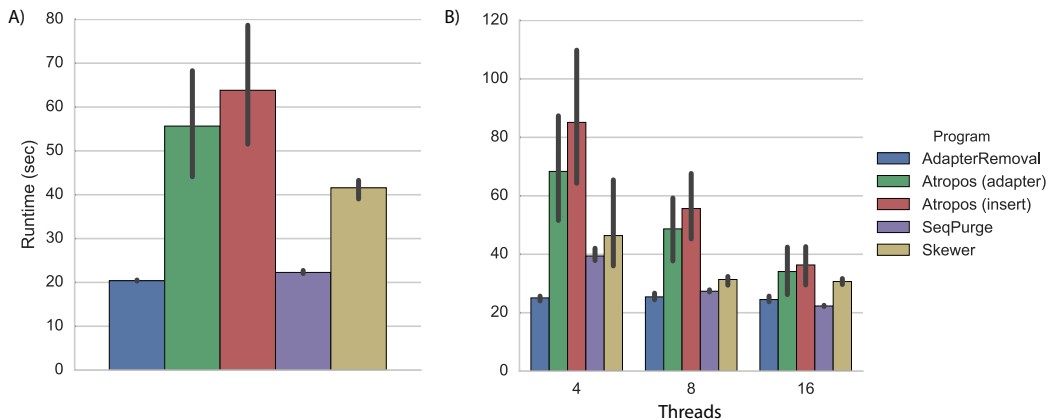

**Figure 2 Trimming execution time for simulated data.** Execution time on simulated datasets for programs running on (A) a desktop computer with four parallel processes (threads), and (B) a cluster node with 4, 8, and 16 threads. Each program was executed multiple times, and Atropos was run with combinations of alignment algorithm (insert-match or adapter-match) and output mode (worker-compression, writer-compression, or parallel-write). The mean execution times for each program are shown with 95% CI.

adapter-match algorithm in parallel-write mode is the fastest of the tools, and with 16 threads Atropos using the insert-match algorithm in parallel-write mode is also faster than the other three tools (Table S3).

Atropos uses substantially more memory than the other tools (Fig. S1; Table S4). We expect this is partially due to overhead of automatic memory management in Python compared to C++ (in which AdapterRemoval, SeqPurge, Skewer are implemented), but in larger part results from Atropos' use of in-memory queues to communicate between parallel processes. For all four programs, memory usage increases slightly with increasing number of threads. We note that Atropos provides parameters to limit memory usage (although typically at the expense of reduced speed).

For most datasets and thread counts, Atropos and Skewer typically achieve the highest mean CPU utilization, indicating that they are less I/O-bound than AdapterRemoval or SeqPurge (Fig. S2).

### Accuracy

We found that the four trimming algorithms had different biases toward under- and over-trimming (Table 2). Across the three sequencing error rates, Skewer had the lowest frequency of wrongly trimming reads while AdapterRemoval had the highest. The Atropos adapter-match algorithm exhibited almost no over-trimming of reads, but also had a very high frequency of under-trimming. The Atropos insert-match algorithm and SeqPurge had similarly low frequencies of under-trimming reads. Overall, the Atropos insert-match algorithm demonstrated the lowest error rates at the read level (0.01%).

In terms of numbers of over- and under-trimmed bases, the Atropos insert-match algorithm and SeqPurge clearly had the best performance (Table 2) at all sequencing error rates. The two algorithms had similarly low numbers of under-trimmed bases, but the

**Table 2 Trimming accuracy on simulated data with three different base-call error rates.**

| Program | Reads | | | | Bases | | |
|---|---|---|---|---|---|---|---|
| | Wrongly trimmed | Over-trimmed | Under-trimmed | Total error (%) | Over-trimmed | Under-trimmed | Total error (%) |
| **Error rate 0.2%** | | | | | | | |
| AdapterRemoval | 664 (0.09%) | 29 (0.00%) | 65 (0.01%) | 0.10 | 6,043 | 2,511 | 0.005 |
| Atropos (adapter) | 51 (0.01%) | **1 (0.00%)** | 28,991 (3.77%) | 3.78 | 490 | 102,133 | 0.057 |
| Atropos (insert) | 60 (0.01%) | 24 (0.00%) | **31 (0.00%)** | **0.01** | 186 | **94** | **0.000** |
| SeqPurge | 94 (0.01%) | 24 (0.00%) | **31 (0.00%)** | 0.02 | 1,574 | **94** | 0.001 |
| Skewer | **18 (0.00%)** | 13 (0.00%) | 146 (0.02%) | 0.02 | **39** | 8,309 | 0.005 |
| **Error rate 0.6%** | | | | | | | |
| AdapterRemoval | 666 (0.09%) | 19 (0.00%) | 69 (0.01%) | 0.10 | 5,547 | 2,032 | 0.004 |
| Atropos (adapter) | 72 (0.01%) | **6 (0.00%)** | 28,843 (3.76%) | 3.77 | 733 | 101,839 | 0.057 |
| Atropos (insert) | 52 (0.01%) | 15 (0.00%) | 42 (0.01%) | **0.01** | 151 | 146 | **0.000** |
| SeqPurge | 78 (0.01%) | 16 (0.00%) | **41 (0.01%)** | 0.02 | 822 | **145** | 0.001 |
| Skewer | **8 (0.00%)** | 8 (0.00%) | 180 (0.02%) | 0.03 | **16** | 11,732 | 0.007 |
| **Error rate 1.2%** | | | | | | | |
| AdapterRemoval | 680 (0.09%) | 16 (0.00%) | 65 (0.01%) | 0.10 | 5,795 | 2,667 | 0.005 |
| Atropos (adapter) | 76 (0.01%) | **5 (0.00%)** | 30,152 (3.92%) | 3.94 | 721 | 117,027 | 0.065 |
| Atropos (insert) | 49 (0.01%) | 13 (0.00%) | **35 (0.00%)** | **0.01** | 111 | **85** | **0.000** |
| SeqPurge | 71 (0.01%) | 13 (0.00%) | **35 (0.00%)** | 0.02 | 1,524 | **85** | 0.001 |
| Skewer | **11 (0.00%)** | 8 (0.00%) | 182 (0.02%) | 0.03 | **19** | 14,261 | 0.008 |

**Note:**
Wrongly trimmed: reads that do not contain adapters but were trimmed anyway; Over-trimmed: reads that contain adapters but from which too many bases were removed; Under-trimmed: reads that contain adapters but from which too few bases were removed. Both read-level and base-level error rates are shown. Fractions of total reads/bases are in parentheses. The best tool(s) in each category is highlighted.

Atropos insert-match algorithm had a lower number of over-trimmed bases, giving it the lowest overall error rate (0.0002%). On the other hand, Skewer and the Atropos adapter-match algorithm left substantial numbers of under-trimmed bases while AdapterRemoval was again biased toward over-trimming.

Additionally, we found that all tools discarded very similar numbers of reads (1.8%) that were below the minimum length threshold of 25 bp after trimming. These were reads with short insert sizes, which have a high rate of spurious mapping, and thus it is common practice to discard them.

## Real data

We first tested Atropos' adapter detection module on the real datasets. Using the first 10,000 reads in each pair of FASTQ files, Atropos correctly detected the exact sequences of the adapters used in constructing each library. For three of the four adapters, the sequences were found in a list of known contaminants (WGBS read 1: "TruSeq Adapter, Index 7"; WGBS read 2 and mRNA-seq read 2: "TruSeq Universal Adapter"); the mRNA-seq read 1 adapter appears to have a custom-designed sequence.

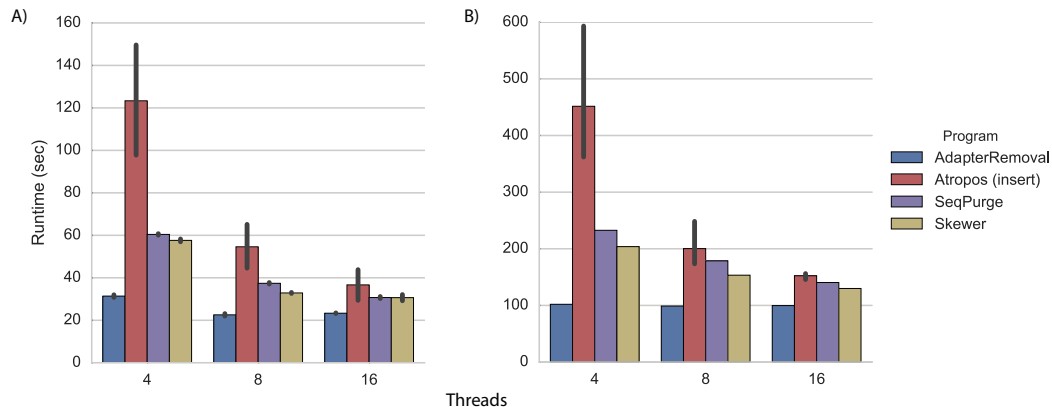

**Figure 3 Trimming execution time for real data.** Execution time on (A) WGBS and (B) mRNA-Seq datasets for programs running on a cluster node with 4, 8, and 16 threads. Each program was executed multiple times, and Atropos was run with the insert-match algorithm and parallel-write output mode. The mean execution times for each program are shown with 95% CI.

## Performance

We performed adapter trimming on the real datasets in the same cluster environment. Again, we found that AdapterRemoval had the fastest execution time (Fig. 3; Tables S5 and S6). When trimming the WGBS data with 16 threads, Atropos (using the insert-match algorithm in parallel-write mode) was nearly as fast as AdapterRemoval (Fig. 3A; Table S5), while on the mRNA-Seq data Skewer, SeqPurge, and Atropos were all 30%–50% slower than AdapterRemoval (Fig. 3B; Table S6).

We also performed read mapping on the cluster with 16 cores. Mapping times were very similar for all algorithms on both the WGBS and mRNA-Seq datasets, and were much faster than for the untrimmed reads (Fig. S3).

## Effectiveness

We assessed read trimming effectiveness in practical terms. For the WGBS data, we computed the number of trimmed reads mapped at a given quality (MAPQ) cutoff, relative to the number of untrimmed reads mapped at that cutoff. We found that trimming by Atropos resulted in the greatest increase in number of mapped reads at all quality cutoffs (Fig. 4A). Trimming with SeqPurge, Skewer, and AdapterRemoval resulted in similar, but smaller, gains in mapping quality. At the highest MAPQ thresholds (45, 50, 55), Atropos substantially outperforms the other three tools.

We also found that additional quality trimming in addition to adapter trimming has a substantial negative effect on read mapping, at least for bisulfite reads mapped using bwa-meth (Fig. 4B). Quality trimming by Skewer had the least negative effect on mapping quality of the four programs, and quality trimming by AdapterRemoval had the greatest negative effect on mapping quality.

For the mRNA-seq data, we additionally compared each alignment to GENCODE (v26) gene annotations (*Harrow et al., 2012*) to determine the number of reads mapped to expressed regions of the genome. We found that trimming with Atropos resulted a greater

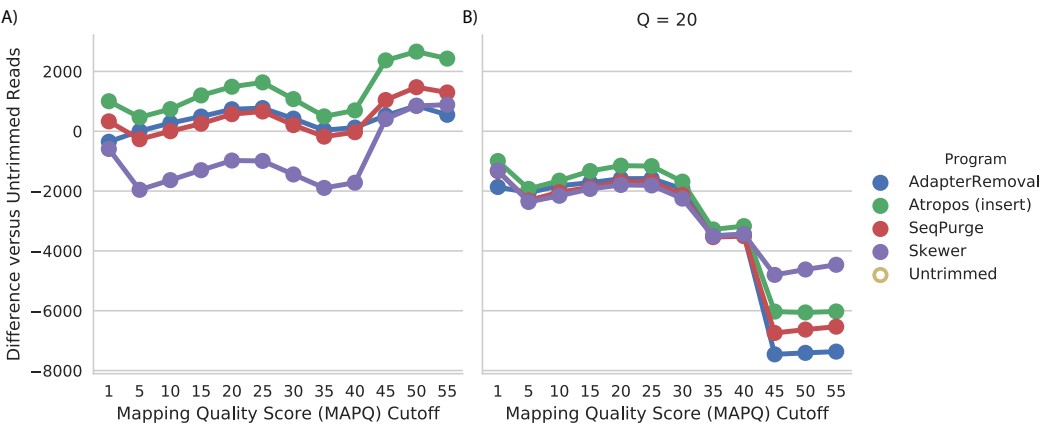

**Figure 4 Atropos trimming best improves mapping of real WGBS sequencing reads.** Reads were adapter-trimmed with all four programs (A) without additional quality trimming ($Q = 0$) and (B) with quality trimming at a threshold of $Q = 20$. We mapped both untrimmed and trimmed reads to the genome. For each MAPQ cutoff $M \in \{0, 5, \ldots, 60\}$ on the x-axis, the number of trimmed reads with MAPQ >= $M$ less the number of untrimmed reads with MAPQ >= $M$ is shown on the y-axis for each program.

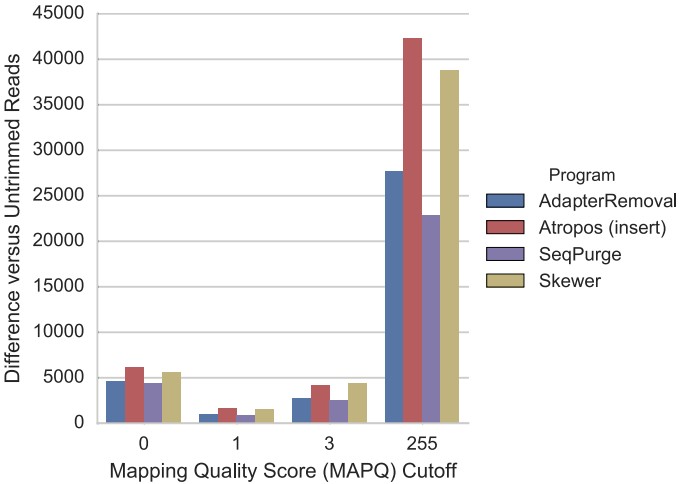

**Figure 5 Atropos trimming results in the greatest increase in mRNA-seq reads mapped to GENCODE regions.** Reads were adapter-trimmed with all four programs without additional quality trimming. We mapped both untrimmed and trimmed reads to the genome using STAR. When parameter `outSAMmultNmax = 2`, STAR produces only four MAPQ values: 255 = unique alignment; 3 = two alignments with similar but unequal score; 1 = two alignments with equal score; and 0 = unmapped. For each MAPQ cutoff $M \in \{0, 1, 3, 255\}$ on the x-axis, the number of trimmed reads that align to GENCODE regions with MAPQ >= $M$ less the number of untrimmed reads with MAPQ >= $M$ is shown on the y-axis for each program.

number of mapped reads aligned to expressed regions compared to the other tools at all MAPQ thresholds (Fig. 5).

## CONCLUSION

Our results demonstrate that adapter trimming tools are approaching optimal accuracy, at least for the (currently) most common type of data—paired-end short reads with 3′ adapters. On synthetic data with varying error rates, Atropos (using our new
insert-match algorithm) and SeqPurge both demonstrated overall error rates of 0.01% at the read level, and Atropos has the lowest base-level error rate of 0.0002%.

On real WGBS and mRNA-seq data, we found that adapter trimming with Atropos resulted in the greatest increase in read mapping quality. We also found that stringent quality trimming has a negative effect on WGBS read mapping quality, at least when using bwa-meth as the alignment tool. For reads trimmed with a quality threshold of 20, all mapping statistics were worse than those for untrimmed reads.

In terms of performance, AdapterRemoval and SeqPurge had the best performance of the four tools tested when only four threads were available, while Atropos had superior performance on the simulated datasets and competitive performance on the real datasets when there were at least eight threads available. Of the three write modes, Atropos performed best in parallel-write mode. However, parallel-write mode has the trade-off of producing a larger number of data files, which may make analyses of large projects more complicated to manage. Atropos' memory requirements were the highest among the four programs (3–4 GB versus 0.5–1.5 Gb), but well within the capabilities of most modern computer systems.

In summary, our results show that Atropos offers the best combination of accuracy and performance of the tools that we evaluated. Furthermore, Atropos has the richest feature set of the four tools, including Methyl-Seq-specific trimming options, automated adapter detection, estimation of sequencing error, computation of quality-control metrics before and after trimming, and support for data generated by many sequencing methods (ABI SOLiD, Illumina NextSeq, mate-pair libraries, and single-end sequencing). Although we have not optimized Atropos for long-read data (e.g., PacBio and Nanopore), it should work on those datasets given appropriate parameter settings, and we plan to soon provide explicit long-read support.

## DATA AVAILABILITY

- The Atropos source code, including detailed instructions and all scripts needed to execute the analyses in this paper, are available at https://github.com/jdidion/atropos. The portions of Atropos developed by JPD are a work of the US government, and thus all copyright is waived under a CC0 1.0 Universal Public Domain Dedication (https://creativecommons.org/publicdomain/zero/1.0/).
- Atropos can be installed using Python 3.3+ and any one of the following methods:
  - From source, using instructions at the aforementioned GitHub repository website.
  - From the Python Package Index (pypi), using the pip tool: "pip install atropos."
  - From the Conda package manager: "conda install atropos."
  - From a Docker container, using a Docker or Singularity client: e.g., "docker run jdidion/atropos."
- The K562 mRNA-seq data (accession SRR521458) is available from the NCBI Sequence Read Archive: https://trace.ncbi.nlm.nih.gov/Traces/sra/?run=SRR521458.

- The GM12878 WGBS data (accession ENCLB794YYH) is available from the ENCODE project website: https://www.encodeproject.org/experiments/ENCSR890UQO/.
- We used human reference genomes GRCh37 and GRCh38, downloaded from http://hgdownload.cse.ucsc.edu/downloads.html#human.
- We used GENCODE v26 annotations, downloaded from ftp://ftp.sanger.ac.uk/pub/gencode/Gencode_human/release_26.
- All datasets, including the simulated DNA-Seq reads, have been packaged into Docker containers, and are available in the Docker Hub (https://hub.docker.com/r/jdidion/). Container definitions are available in the aforementioned GitHub repository.

## ACKNOWLEDGEMENTS

We thank Jim Mullikin, Stephen Piccolo, and two anonymous reviewers for helpful comments on an earlier version of this manuscript. We also thank the users of Cutadapt and Atropos that have contributed bug reports and improvements.

### Funding

JPD and FSC are funded by the NIH intramural program. Additionally, JPD is funded by the American Diabetes Association (1-17-PDF-100). MM is supported by a grant from the Knut and Alice Wallenberg Foundation to the Wallenberg Advanced Bioinformatics Infrastructure. The funders had no role in study design, data collection and analysis, decision to publish, or preparation of the manuscript.

### Grant Disclosures

The following grant information was disclosed by the authors:
NIH intramural program.
American Diabetes Association: 1-17-PDF-100.
Knut and Alice Wallenberg Foundation to the Wallenberg Advanced Bioinformatics Infrastructure.

### Competing Interests

The authors declare that they have no competing interests.

### Author Contributions

- John P. Didion conceived and designed the experiments, performed the experiments, analyzed the data, contributed reagents/materials/analysis tools, wrote the paper, prepared figures and/or tables, reviewed drafts of the paper.
- Marcel Martin contributed reagents/materials/analysis tools, reviewed drafts of the paper.
- Francis S. Collins reviewed drafts of the paper.

## Data Availability

GitHub: https://github.com/jdidion/atropos;

Didion J. (2016, September 15). Jdidion/Atropos: Atropos V1.0.15. *Zenodo*
DOI 10.5281/zenodo.154097.

## Supplemental Information

Supplemental information for this article can be found online at http://dx.doi.org/
10.7717/peerj.3720#supplemental-information.

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
