# Peer review of "Atropos: specific, sensitive, and speedy trimming of sequencing reads"

_PeerJ, doi:10.7717/peerj.3720_

## Round 0.1 · original submission · Minor Revisions

The manuscript is nicely written, the results pretty convincing and the methodology is sound. Trimming is an important and often neglected step in NGS pipelines and the method here proposed sound promising. There are some minor concerns raised by the three external reviewers which will need to be carefully addressed before acceptance of the manuscript, including the comparison with other existing methods. Please, be also aware that the manuscript will be sent back to the original reviewers to verify if all the points have been properly solved.

Reviewer 1 ·

Basic reporting

In overall, the manuscript is written in a clear and unambiguous English and the different terms used are consistent. I therefore only have very small (pedantic) things to say in connection with the spelling, but since I found them I think I’ll share them anyway and then you can decide to use time on it or not.


Minor things about the consistency in the spelling of the terms:
Decide whether to use hyphens or not in terms like “writer(-)compression”, “parallel(-)write” and “worker(-)compression” since these are written differently when looking at the tables and the text.

In addition, I found some minor errors when reading the text:
l. 285 ‘of’ used twice.
l. 322 at → a

Lastly, I suggest to remove *-signs from formula 3-4, since these are not used in the other formulas or alternatively use multiplication sign (\cdot in LaTex).


The manuscript has a good introduction explaining why the trimming tools are important and why they can still be developed into faster and more accurate tools. Adapters are assumed to be known by the reader. Overall, all figures and tables presented in this text seems to be relevant in connection with the aims defined. They are generally well labeled and described. With that said, here is my comments for further improvement in prioritized order:

Last sentence of description text for figure 2-3 is a bit confusing to read and understand. I suggest reformulating these and in addition to use >= signs instead of “greater than or equal to”.
MAPQ differences in Figure 2 and 3. MAPQ in Fig. 3 seems to be more a qualitative score than in fig 2. Does it make sense to make a curve plot for the qualitative score, since there are no values in between? Furthermore, what tool/method is used for mapping WGBS seq. reads to the genome?
The caption for table 4 only contains one line and needs to be elaborated. I suggest: Explain if the Overtrimmed in bases is both bases for “over trimmed” and “wrongly trimmed”, which it seems like when looking at the numbers, but is not obvious. Describe the percent in parenthesis (total number of reads I guess). Describe the different error rates and notable results highlighted in bold.
In the caption of table 2, the worker-compression is missing when describing the combinations of Atropos.
Use capital letters for the different tools in the legend for figure 3.
Table 3: Not crucial, but I think it would make more sense to shift the order of the two titles; “Execution time” and “4 Threads 8 Threads 16 Threads” so that “Execution time” comes after “4 Threads..”, since “4 Threads..” is the “global” title and “Execution time” is the “local” title of the table.

The manuscript has a thorough description of the data used and how to obtain and reproduce the data results. These includes:

Link to source code and scripts needed to execute analysis of the paper
link to simulated and real data
Link to the human reference genome used
Link to GENCODE annotations


It also contains necessary information for easy installation of Atropos, assuming that the reader knows about the pip tool in Python.

Experimental design

Overall, the manuscript defines three relevant aims in the introduction for improving an already available tool. The improvements of the two first aims are measured by different analyses which tries to capture the time & memory consumption as well as the accuracy of the tool compared with other tools. Measuring the amount of “wronly trimmed“, “overtrimmed” and “undertrimmed” as well as comparing the amount of trimmed versus untrimmed reads for specific MAPQ scores seems like a coherent way of getting a grasp of the different tools accuracy in connection with identifying and trimming out read containments.

Validity of the findings

In overall I find the validity of this manuscript compelling. The manuscript has used tools for comparisons that have already been marked as one of the best tools available. Furthermore the different measurements for comparing the different tools which are easy to understand and give insights in both time & memory as well as accuracy of the tools in question.
One concern I have about the data in connection with the validity in the manuscript is the use of the older GRCh37 in stead of the newest GRCh38 reference genome. Since the manuscript tries to measure mapping differences between trimmed and untrimmed reads on real data, I think there is some motivation of always using the newest and most reliable genome reference consortium available. Therefore it could be interesting to see the same analysis done for GRCh38 data to confirm the results obtained in this manuscript.

·

Basic reporting

The paper is well written. The authors did a nice job of comparing their method in a comprehensive way against the latest competing tools. I also like that their software builds on a current, commonly used tool so that others will be able to adapt to it (no pun intended) more easily. Below I have noted a few minor suggestions for improving the text.

* In the Abstract, it says, "makes it an appropriate choice for the pre-processing of most current- generation sequencing data sets." It would be helpful to state more specifically what types of sequencing data for which this method is suitable.
* It would be slightly more clear if you changed "(as it is currently only able to use a single processor)" on line 59 to "(as Cutadapt is currently only able to use a single processor)". When reading it the first time, I had to think twice about whether you were referring to Atropos or Cutadapt there.
* Figure 1 is really helpful.
* On line 81, change "as a intuitive parameter" to "as an intuitive parameter".
* "Let s be the adapter, t the read and m = |s|, n = |t|." It would be more intuitive to use a variable called "a" to represent the adapter and "r" the reader. This may be due to my limited math background, but I am not sure I understand how s is represented. Is it a numeric value? How is that number obtained from the adapter?
* On line 244, it says, "the fragment is under-trimmed, or the fragment is over-trimmed." I assume over-trimmed is distinct from wrongly trimmed in that the fragment does contain an adapter but the tool trims more than the adapter sequence. It would be helpful to clarify this, just to make sure readers understand the distinction.
* On line 316, skewer should be capitalized.

Experimental design

* The authors use bisulfite sequencing and mRNA sequencing data. But there's no real-world DNA-Sequencing data. This type of data would be helpful to include because this type of data is so common. You could potentially use the Genome in a Bottle data to compare the trimming methods and see which leads to more accurate identification of known DNA variants.

Validity of the findings

* I love the adapter detection feature. I am not quite sure I understood how the user can avoid false positives when using this feature. It would be helpful to add some type of explanation about this.
* The parallelization feature is very nice. It was great that the authors showed how the performance gains improve with an increasing number of threads. This is just personal preference, but it would be more intuitive if Table 3 were represented as a figure rather than a table.
* The comparisons against other state-of-the-art trimming tools are appreciated. It is helpful to see benefits and limitations of each tool.

Additional comments

* It is nice that you can install this tool via pip. However, there are several dependencies to install. Importantly, this tool only runs on Python 3.3+. For people (like me) who are still using Python 2, it is a barrier to install 3.3+ and have two versions. It would be convenient if this software and its dependencies were packaged in a Docker container so that it would be easier to install but also so that I would not be required to install Python 3.3+. It should be not much work to build the Docker image and store it on DockerHub.com. Below I have pasted the contents of the Dockerfile that I used to test your software.
* It's fantastic that the authors provide their analysis scripts and tests in their GitHub repository. I did have trouble accessing the simulated data in the GitHub repository. I cloned the repository. But it only had a short description of the FASTQ files. It seems there is a special way to download these larger files, but I wasn't sure how to do that. Providing a README would be helpful.

FROM python:3.5

COPY SRR1972917_*.fastq /

RUN pip install atropos
RUN git clone https://github.com/jdidion/atropos.git
RUN atropos --threads 2 -a AGATCGGAAGAGCACACGTCTGAACTCCAGTCACGAGTTA -o trimmed1.fq.gz -p trimmed2.fq.gz -pe1 /SRR1972917_1.fastq -pe2 /SRR1972917_2.fastq

Reviewer 3 ·

Basic reporting

The paper is well written and easy to follow.

Experimental design

The authors selected existing adapter trimming software for comparison based on the benchmarks presented in Sturm et al., 2016. However, I note that Sturm et al., 2016 made use of a version of AdapterRemoval (1.5.4) that was already greatly outdated when that paper was published. I would be curious as to how Atropos performs in comparison to a more recent version of AdapterRemoval (2.x), which greatly outperform the version tested in Sturm et al., 2016 and implements multi-threaded operation:
http://bmcresnotes.biomedcentral.com/articles/10.1186/s13104-016-1900-2

Secondly, the authors benchmark the mapping runtime following trimming, but do not specify what versions of the mapping software, and what parameters (if any), they used to carry out this mapping. Nor could I find any mention of the exact number of times that programs were executed to estimate the minimum and maximum runtimes.

Validity of the findings

no comment

Additional comments

I would suggest that the authors specify 'Python 3' on lines 72, rather than just 'Python'. This is a very minor issue, and is included simply because my first attempt at installing Atropos failed due to using the Python 2 version of pip.

---

## Round 0.2 · accepted · Accept

All the previous concerns have been nicely addressed. We are now glad to endorse this manuscript for publication.

·

Basic reporting

No additional comments.

Experimental design

No additional comments.

Validity of the findings

No additional comments.

Additional comments

I am content with the changes that have been made. I feel that no additional changes are needed.